# Quantum Switchboard with Coupled-Cavity Array

**DOI:** 10.3390/e24010136

**Published:** 2022-01-17

**Authors:** Wai-Keong Mok, Leong-Chuan Kwek

**Affiliations:** 1Centre for Quantum Technologies, National University of Singapore, Singapore 117543, Singapore; waikeong_mok@u.nus.edu; 2MajuLab, CNRS-UNS-NUS-NTU International Joint Research Unit, Singapore UMI 3654, Singapore; 3National Institute of Education, Nanyang Technological University, Singapore 637616, Singapore; 4Quantum Science and Engineering Center, Nanyang Technological University, Singapore 637616, Singapore

**Keywords:** quantum entanglement, quantum network, coupled-cavity array

## Abstract

The ability to control the flow of quantum information is deterministically useful for scaling up quantum computation. In this paper, we demonstrate a controllable quantum switchboard which directs the teleportation protocol to one of two targets, fully dependent on the sender’s choice. Importantly, the quantum switchboard also acts as a optimal quantum cloning machine, which allows the receivers to recover the unknown quantum state with a maximal fidelity of $\frac{5}{6}$. This protects the system from the complete loss of quantum information in the event that the teleportation protocol fails. We also provide an experimentally feasible physical implementation of the proposal using a coupled-cavity array. The proposed switchboard can be utilized for the efficient routing of quantum information in a large quantum network.

## 1. Introduction

A quantum network contains many quantum nodes for processing and storing quantum states and quantum channels for the distribution of quantum information [1,2,3]. The ability to distribute arbitrary quantum states is essential for quantum information processing in a quantum network. Efficient navigation in a complex quantum network will eventually need a quantum multiplexer or switchboard to direct the flow of quantum information. A likely scenario is one where we have many quantum devices or computers linked by a switchboard system to other devices and computers for processing, storing or sensing (see Figure 1). Many quantum information processes require the explicit preparation of specially entangled quantum states. Two-qubit maximally entangled state often called Bell state, for instance, form an essential quantum resource needed in quantum teleportation [4]. The preparation of three-qubit maximally entangled state (such as GHZ state) could be harnessed for secure secret sharing [5]. In one-way quantum computing, a four-qubit entangled state called cluster state provides an efficient implementation of a universal quantum gate: arbitrary single-qubit unitary operation [6].

Entangled states which are used as a common resource in quantum information processes generally need not even be maximally entangled at all. As long as the state is genuinely entangled, quantum computation and communication will generally be better than the classical counterparts. In particular, the nonmaximally entangled W state was experimentally implemented and proposed for controlled quantum teleportation and secure communication [7].

An essential component of any quantum computation is the ability to spread quantum information over various parts of the quantum computer. The parts then undergo separate evolutions depending on the type of the quantum information processing we wish to implement. Ultimately, we need to be capable of navigating the relevant part of the information into a designated output. In a classical computer, this flow of information is achieved through a controllable switch. Is it possible to design a quantum analogue for such a device? An added complexity in a quantum switch would be the requirement that the information flows down many possible channels coherently as well as the possibility of channeling it in one selected direction.

Ideally, we would like to realize a simple device to achieve this purpose. In addition, these qubits are implemented in practice in a physical system determined by the nature of the qubits and couplings between them. Therefore, in the design of the switch, realistic interaction between the qubits severely limits the type of possible Hamiltonians that can execute such a quantum switch. Here, we present a possible implementation of the switch that minimally fulfills these requirements.

In the usual teleportation protocol [4], the sender Alice and receiver Bob begins by sharing a two-qubit maximally entangled Bell state. Alice also possesses an additional ancilla qubit which carries the quantum state to be teleported. Alice then performs a Bell measurement on her share of the entangled state and the ancilla qubit, and the measurement results are then communicated classically to Bob which will perform a corresponding unitary operation on his qubit to recover the desired state, thus completing the teleportation. The choice of receiver is fixed by whoever Alice shares the initial entangled state with, and cannot be changed without discarding the entangled state and setting up a new one with a different receiver Charlene. This poses two limitations to the scheme: to teleport quantum states on-demand with multiple receivers, Alice has to share a separate entangled qubit pair with each of the receivers, which must be isolated from one another. Furthermore, Alice is not allowed to choose the receiver after the Bell measurement in this scheme.

In this paper, we show how the above challenges can be overcome by using a specific four-qubit entangled state shared between the sender Alice, the two receivers Bob and Charlene, as well as the accomplice Dick. The role of Dick is to direct the teleported state to the desired receiver as per the sender’s choice, which can be done after Alice reveals the Bell measurement results. By controlling the flow of quantum information deterministically, the protocol becomes a quantum switchboard with Dick acting as a ‘switchboard operator’. Previous proposals of controlled quantum teleportation [8,9,10,11,12,13,14,15] share the common problem of significant loss of quantum information in the event of teleportation failure, which in the worst-case scenario causes the receiver qubit to be in a maximally-mixed state indicating a complete information loss. The teleportation protocol can fail if, for example, the classical communication channel is attacked which hinders Alice from announcing the results of her Bell measurement. Our proposed switchboard has the unique feature of also being an optimal quantum cloning machine, which allows the receivers to recover the unknown quantum state with the maximal fidelity of $\frac{5}{6}$ even if the teleportation protocol fails. We also provide an experimentally feasible physical implementation of the quantum switchboard using a driven coupled cavity array [16], which realizes the required multipartite entangled state.

## 2. Controllable Quantum Switchboard

The controllable quantum switchboard serves to direct the flow of quantum information in a quantum network [17]. Specifically, it allows Alice to transfer her quantum state perfectly to either Bob or Charlene (free choice) with the help of her accomplice Dick. The basis of the scheme requires all four parties to be initialized in the four-qubit entangled state
(1)$$\begin{array}{cc} |\psi \rangle & =\frac{1}{\sqrt{3}}\left(|{\left(11\right)}_{AB}\rangle |{\left(11\right)}_{DC}\rangle -|{\left(11\right)}_{AC}\rangle |{\left(11\right)}_{BD}\rangle \right)\\ & =\frac{1}{2\sqrt{3}}{\left(2|0110\rangle +2|1001\rangle -|0101\rangle -|1010\rangle -|0011\rangle -|1100\rangle \right)}_{ABCD} \end{array}$$
using the notation for Bell states
(2)$$|\left(ab\right)\rangle =\sum_{k=0}^{1}\frac{{\left(-1\right)}^{kb}}{\sqrt{2}}|k,k⊕a\rangle$$
with ⊕ denoting modulo-2 addition, or an XOR operation. For example, the state |(11)〉 represents the singlet state $\frac{1}{\sqrt{2}}\left(|0\rangle |1\rangle -|1\rangle |0\rangle \right)$. It turns out that this state is ideally suited for a quantum switchboard, i.e., a circuit that can be used to direct the flow of quantum information in a controllable manner. An interesting property of the presented switchboard is that in the case of failure the information is not entirely lost.

We will now illustrate the quantum switchboard protocol. Suppose Alice wants to teleport her quantum state ${|\alpha \rangle }_{0}$ encoded in an ancilla qubit (indexed 0) to Bob. The combined state of the ancilla and the initial entangled state can be expressed as
(3)$${|\alpha \rangle }_{0}|\psi \rangle =\frac{1}{4\sqrt{3}}\sum_{k,l,m,n=0}^{1}\lambda_{kl}|{\left(mn\right)}_{0A}\rangle ⊗U_{mn,kl}{|\alpha \rangle }_{B}|{\left(kl\right)}_{DC}\rangle$$
where $\lambda_{11}=3$, $\lambda_{01}=\lambda_{10}=1$ and $\lambda_{00}=-1$. $U_{mn,kl}$ is the unitary transformation on Bob’s qubit, determined by the results of both Bell measurements (mn) and (kl). Explicitly, the set of unitaries $U_{mn,kl}$ are the Pauli operators {I,X,Y,Z} where $U_{mn,kl}=I$ if m⊕k=n⊕l=0, *X* if m⊕k=1 and n⊕l=0, *Y* if m⊕k=n⊕l=1, and lastly *Z* if m⊕k=0 and n⊕l=1. Alice first performs a Bell measurement on the ancilla and her share of |ψ〉, yielding a measurement result (mn). Immediately after getting one of the four possible outcomes, she broadcasts two (classical) bits of information to Bob and Charlene as it is in the usual teleportation scheme. At this point, it is not necessary for Dick to know these two bits of information.

The Bell measurement collapses the state of the other three parties to be
(4)$$|\chi_{mn}\rangle =\sum_{k,l=0}^{1}\frac{\lambda_{kl}}{2\sqrt{3}}U_{mn,kl}{|\alpha \rangle }_{B}|{\left(kl\right)}_{DC}\rangle$$

If we trace out Charlene and Dick’s qubits, then Bob will have the mixed state
(5)$$\rho_{mn}=\frac{1}{12}\sum_{k,l=0}^{1}{\left|\lambda_{kl}\right|}^{2}U_{mn,kl}|\alpha \rangle \langle \alpha |U_{mn,kl}^{†}$$

A similar state is obtained for Charlene if we trace out Bob and Dick’s qubits instead. For example, if the Bell measurement yields (11), Bob’s state becomes
(6)$$\rho_{11}=\frac{1}{12}\left(9|\alpha \rangle \langle \alpha |+X|\alpha \rangle \langle \alpha |X+Y|\alpha \rangle \langle \alpha |Y+Z|\alpha \rangle \langle \alpha |Z\right)$$

The fidelity of the state recovered by Bob is $F_{\alpha }=\langle \alpha |\rho_{11}|\alpha \rangle =\frac{3}{4}+\frac{1}{12}{\left(|\langle X\rangle \right|}^{2}+{\left|\langle Y\rangle \right|}^{2}+{\left|\langle Y\rangle \right|}^{2})$. Writing the arbitrary state as |α〉=cos(θ/2)|0〉+exp(iϕ)sin(θ/2)|1〉 and averaging over the Bloch sphere, we have
(7)$$\frac{1}{4\pi }\int_{0}^{2\pi }d\phi \int_{0}^{\pi }d\theta \sin\theta {\left|\langle \alpha |\hat{O}|\alpha \rangle \right|}^{2}=\frac{1}{3},\;\hat{O}=X,Y,Z$$

Hence, Bob and Charlene can recover the state of the ancilla qubit with a fidelity of $\frac{5}{6}$. There is no loss of generality in assuming the Bell measurement result to be (11) since the state in Equation (Equation 6) can be obtained for any (mn) by simply performing an additional unitary transformation $U_{mn,11}^{†}$ on the state $\rho_{mn}$ based on the knowledge of the broadcast classical bits. Equivalently, we can also write the state in Equation (Equation 6) as $\frac{5}{6}|\alpha \rangle \langle \alpha |+\frac{1}{6}|\alpha^{⊥}\rangle \langle \alpha^{⊥}|$, where $|\alpha^{⊥}\rangle$ is the state orthogonal to |α〉, from which the cloning fidelity of $\frac{5}{6}$ becomes apparent. The given state at the beginning does not provide a universal cloning machine for three copies of the cloned state [18]. In fact, tracing out Bob and Charlene’s qubits after the Bell measurement, and performing an appropriate unitary transform on Dick’s qubit depending on the measurement outcome, we can write Dick’s qubit as $\rho_{D}=\frac{1}{3}|\alpha \rangle \langle \alpha |+\frac{2}{3}|\alpha^{⊥}\rangle \langle \alpha^{⊥}|$. Thus, the qubit belonging to Dick is related to the Alice’s ancilla qubit with the “classical” fidelity $\frac{1}{3}$, i.e., the fidelity that can be achieved without prior entanglement. Interestingly, Bob and Charlene possess the optimum fidelity achievable under a symmetric cloning machine. Dick’s fidelity is allowed since there is no limitation on the production of clones with the fidelity below $\frac{2}{3}$ using a trivial cloning scheme, such as by measuring the qubit on a random basis [18].

To obtain perfect fidelity, the accomplice Dick can send his qubit to Charlene to perform Bell measurement, yielding the result (kl). With both measurement results (mn) from Alice and (kl) from Charlene, Bob now has the state
(8)$${|\phi \rangle }_{B}=U_{mn,kl}|\alpha \rangle$$

Thus, Bob can recover Alice’s state |α〉 (up to a global phase) by simply performing an inverse transform $U_{mn,kl}$ on his qubit. The presented protocol behaves like an optimal telecloner [19]. However, there is still an unused qubit held by Dick. Depending on Alice’s decision regarding to whom she wishes ultimately to send her auxiliary qubit; say Bob (Charlene) for instance, she can direct Dick to send his qubit to Charlene (Bob). As soon as Charlene receives Dick’s qubit, he can perform a Bell measurement on his qubit with Dick’s qubit and send the results of his measurement to Bob. Using the information from Charlene, Bob can perfectly recover the state of the Alice’s auxiliary qubit. Alternatively, Dick can also make the decision on whom he wishes to transmit the unknown qubit held originally by Alice.

The situation is entirely symmetric, i.e., Dick can send his qubit to Bob instead of Charlene with the result that now Charlene can obtain Alice’s auxiliary qubit with perfect fidelity. In short, the state acts as a quantum switchboard in which Alice can direct optimal clones to Bob and Charlene or perform perfect quantum teleportation to Bob or Charlene by utilizing Dick’s qubit as in a quantum demultiplexer. A schematic diagram of this quantum switchboard protocol is shown in Figure 2. By directing Dick’s qubit to either Bob (or Charlene), Alice can effectively transfer the unknown auxiliary qubit to Charlene (or Bob). Moreover, she can delay the transfer process to a later time as long as she has effective control over Dick’s qubit. The relative phase of π between the components of the state |ψ〉 is crucial for desired functionality. Other phase choices or, for that matter, the complete lack of coherence, will not give us the same quantum switch. To show this, we can consider adding a relative phase $e^{i\phi }$ between the two components. By a similar calculation as before, we find that the cloning fidelity is now $\frac{3-2\cos\phi }{2(2-\cos\phi )}$, which gives the maximal value of $\frac{5}{6}$ only for ϕ=π(mod2π).

Also, if we initialize the qubits in singlet pairs, i.e., the first term in Equation (Equation 1), the deterministic controlled teleportation to either Bob or Charlene can be achieved trivially. However, tracing out Alice, Bob, and Dick’s qubits, we find that Charlene’s qubit is always in the maximally mixed state $\frac{1}{2}I$ which implies that the quantum information is entirely lost. Hence, our proposed initial state is necessary to achieve the dual functionality of both optimal quantum cloning and a deterministic teleportation protocol. Furthermore, in an actual network of nodes, the scheme is easily extendable to a Bethe lattice of nodes within the switchboard configuration as shown in Figure 3. Naturally, the configuration shown in the figure is not the only possibility.

Incidentally, replacing the four-qubit state with a GHZ state, shared among Alice, Bob, and Charlene, one could in principle provide perfect quantum teleportation to both Bob and Charlene without the additional benefit of an optimal quantum cloner [8]. In this case, Alice teleclones to both Bob and Charlene with a classical fidelity of $\frac{2}{3}$. The eventual quantum teleportation to Bob (or Charlene) is performed with a measurement in the basis $\frac{1}{\sqrt{2}}(|0\rangle \pm \left|1\rangle \right)$. The presence of singlets or dimer-like bonds in the four-qubit state renders it more insensitive to perturbation in the quantum critical regimes. This latter feature is absent in the GHZ state.

## 3. Realization of Next Nearest-Neighbor Interactions with Coupled-Cavity Array

Remarkably, the Majumdar–Ghosh (MG) model given by the spin-chain Hamiltonian [20]
(9)$$H_{MG}=J\sum_{i=1}^{N}\left(2{\vec{S}}_{i}{\vec{S}}_{i+1}+\alpha {\vec{S}}_{i}{\vec{S}}_{i+2}\right)$$
and periodic boundary conditions is exactly solvable for α=1 and even number of spins, with the degenerate ground state manifold spanned by the basis (dimer) states
(10)$$\begin{array}{cc} |\psi_{g}\rangle & =|{\left(11\right)}_{12}\rangle |{\left(11\right)}_{34}\rangle \ldots |{\left(11\right)}_{(N-1)N}\rangle \\ |\phi_{g}\rangle & =|{\left(11\right)}_{23}\rangle |{\left(11\right)}_{45}\rangle \ldots |{\left(11\right)}_{N1}\rangle \end{array}$$

Thus, the required four-qubit state in Equation (Equation 1) can be prepared from the superposition of the Majumdar–Ghosh Hamiltonian ground states. While the Majumdar–Ghosh model is yet to be realized experimentally, close proximity to the Majumdar–Ghosh point with α=0.9 was experimentally observed in the Cu${}^{2+}$ mineral szenicsite Cu${}_{3}$(MoO${}_{4}$)(OH)${}_{4}$ [21], showing the dimerized phase. We also note that both the local and non-local dimers in Equation (Equation 10) can be prepared using the dark states of a driven-dissipative spin chain coupled indirectly via mediating photons in a chiral waveguide [22,23]. In this waveguide quantum electrodynamics (QED) setup, the infinite-range nature of the effective spin-spin interactions give rise to interesting dimer states depending on the detuning pattern of the spins and the chirality of the waveguide, which are by design robust against decoherence effects in the waveguide. This means that the controllable quantum switchboard can also be potentially realized with a cold-atom chain. Additionally, chiral waveguide QED realizes the paradigm of cascaded systems, which was exploited in several proposals for long-distance quantum information processing [24,25].

In this section, we provide a different physical implementation using a coupled-cavity array. We directly engineer the Majumdar–Ghosh Hamiltonian which yields the desired switchboard state as the ground state. We review the proposal introduced in Ref. [26] to realize next nearest neighbor (NNN) spin-chain interactions using a coupled-cavity array, and show how it can be used to implement the quantum switchboard. Consider an array of *N* cavity QED subsystems. Each subsystem comprises a four-level atom coupled to a bimodal cavity, with 6 optical driving fields. This forms an effective double Λ system, with $\Lambda_{a}=\left\{|1\rangle ,|2\rangle ,|3\rangle \right\}$ and $\Lambda_{b}=\left\{|1\rangle ,|2\rangle ,|4\rangle \right\}$ (as shown in Figure 4). The ground states |1〉 and |2〉 will be used as a logical qubit.

At the *j*th site, the atom is coupled to 2 cavities (modes $a_{j}$ at frequency $\omega_{a}$ and $b_{j}$ at frequency $\omega_{b}$, with coupling strengths $g_{a}$ and $g_{b}$ respectively). The four driving lasers have Rabi frequencies $\Omega_{1},\Omega_{2}e^{-i\pi j},\Omega_{3}$ and $\Omega_{4}e^{-i\pi j}$, with laser frequencies $\omega_{1},\omega_{2},\omega_{3}$ and $\omega_{4}$ respectively. The transitions are labelled in Figure 4. We can define the detunings $\delta_{31}=\omega_{31}-\omega_{a}$, $\delta_{42}=\omega_{42}-\omega_{b}$, $\Delta_{31}=\omega_{31}-\omega_{1}$, $\Delta_{42}=\omega_{42}-\omega_{2}$, $\Delta_{32}=\omega_{32}-\omega_{3}$ and $\Delta_{41}=\omega_{41}-\omega_{4}$.

Assuming the detunings to be large compared to the Rabi frequencies $|g_{a}\left|,\right|g_{b}|,\Omega_{i}$, we can apply the rotating wave approximation and adiabatically eliminate the excited states |3〉 and |4〉. This leads to the Hamiltonian
(11)$$\begin{array}{cc} H & =-\sum_{j}\left[A_{1}|1_{j}\rangle \langle 1_{j}|a_{j}e^{i\delta_{1}t}+A_{2}|2_{j}\rangle \langle 1_{j}|a_{j}e^{i\delta_{2}t}+H.\;c.\right]\\ & -\sum_{j}\left[{\left(-1\right)}^{j}B_{1}|2_{j}\rangle \langle 2_{j}|b_{j}e^{i\delta_{1}t}+{\left(-1\right)}^{j}B_{2}|1_{j}\rangle \langle 2_{j}|b_{j}e^{i\delta_{2}t}+H.\;c.\right]\\ \; & -\sum_{j}\left[A_{3}|1_{j}\rangle \langle 2_{j}|e^{i\delta_{3}t}+B_{3}|2_{j}\rangle \langle 1_{j}|e^{i\delta 3t}+H.\;c.\right]\\ & -\sum_{j}\left[\left(\frac{\Omega_{1}^{2}}{\Delta_{31}}+\frac{\Omega_{4}^{2}}{\Delta_{41}}\right)|1_{j}\rangle \langle 1_{j}|+\left(\frac{\Omega_{3}^{2}}{\Delta_{32}}+\frac{\Omega_{2}^{2}}{\Delta_{42}}\right)|2_{j}\rangle \langle 2_{j}|\right]\\ \; & -\sum_{j}\left[\frac{g_{a}^{2}}{\delta_{31}}|1_{j}\rangle \langle 1_{j}|a_{j}^{†}a_{j}+\frac{g_{b}^{2}}{\delta_{42}}|2_{j}\rangle \langle 2_{j}|b_{j}^{†}b_{j}\right]\\ & +\sum_{j}\left[J_{a}\left(a_{j}^{†}a_{j+1}+a_{j}a_{j+1}^{†}\right)+J_{b}\left(b_{j}^{†}b_{j+1}+b_{j}b_{j+1}^{†}\right)\right] \end{array}$$
where the coupling parameters are given by
(12)$$\begin{array}{cc} A_{1} & =\frac{\Omega_{1}g_{a}}{2}\left(\frac{1}{\Delta_{31}}+\frac{1}{\delta_{31}}\right),\;A_{2}=\frac{\Omega_{3}g_{a}}{2}\left(\frac{1}{\Delta_{32}}+\frac{1}{\delta_{32}}\right)\\ A_{3} & =\frac{\Omega_{1}\Omega_{3}}{2}\left(\frac{1}{\Delta_{31}}+\frac{1}{\Delta_{32}}\right),\;B_{1}=\frac{\Omega_{2}g_{b}}{2}\left(\frac{1}{\Delta_{42}}+\frac{1}{\delta_{42}}\right)\\ B_{2} & =\frac{\Omega_{4}g_{b}}{2}\left(\frac{1}{\Delta_{41}}+\frac{1}{\delta_{42}}\right),\;B_{3}=\frac{\Omega_{2}\Omega_{4}}{2}\left(\frac{1}{\Delta_{41}}+\frac{1}{\Delta_{42}}\right) \end{array}$$
the two-photon detunings are defined as $\delta_{1}=\delta_{31}-\Delta_{31}=\delta_{42}-\Delta_{42}$, $\delta_{2}=\delta_{31}-\Delta_{32}=\delta_{42}-\Delta_{41}$ and $\delta_{3}=\delta_{1}-\delta_{2}$. $J_{a}$ and $J_{b}$ are the tunneling rates for the NN hopping term between the cavities. The ${\left(-1\right)}^{j}$ factors come from the choice of coupling phases in $\Omega_{2}$ and $\Omega_{4}$. The terms containing $A_{i}$ and $B_{i}$ originate from $\Lambda_{a}$ and $\Lambda_{b}$ respectively (defined in the caption of Figure 4). Using the ground states $|1_{j}\rangle$ and $|2_{j}\rangle$ as qubits, we can define the spin operators $S_{j}^{Z}=\frac{1}{2}\left(|2_{j}\rangle \langle 2_{j}|-|1_{j}\rangle \langle 1_{j}|\right),S_{j}^{+}=|2_{j}\rangle \langle 1_{j}|,S_{j}^{-}=|1_{j}\rangle \langle 2_{j}|$.

To simplify the Hamiltonian, we first observe that the third and fourth terms contribute an effective local magnetic field $h_{j}S_{j}^{Z}$ and can be temporarily ignored. The fifth term modifies the two-photon detunings, giving $\delta_{a1}=\delta_{1}+g_{a}^{2}/\delta_{31}$, $\delta_{a2}=\delta_{2}+g_{a}^{2}/\delta_{31}$, $\delta_{b1}=\delta_{1}+g_{b}^{2}/\delta_{42}$ and $\delta_{b2}=\delta_{2}+g_{b}^{2}/\delta_{42}$. Assuming periodic boundary conditions, we can take the Fourier transform of $a_{j}$ and $b_{j}$:(13)$$a_{j}\left(b_{j}\right)=\frac{1}{\sqrt{N}}\sum_{k=1}^{N}F_{jk}c_{k}\left(d_{k}\right),\;F_{jk}=\exp\left(-i\frac{2\pi }{N}jk\right),\;\sum_{j=1}^{N}F_{jk}^{*}F_{jl}=N{\tilde{\delta }}_{kl}$$
where ${\tilde{\delta }}_{kl}$ is the Kronecker delta. Going into the rotating frame and assuming large detunings, we can adiabatically eliminate the cavities and obtain (after some algebra)
(14)$$\begin{array}{cc} H & =\sum_{j=1}^{N}2\left(\frac{J_{a}}{\delta_{a1}^{2}}A_{1}^{2}-\frac{J_{b}}{\delta_{b1}^{2}}B_{1}^{2}\right)S_{j}^{Z}S_{j+1}^{Z}+\left(\frac{J_{a}}{\delta_{a2}^{2}}A_{2}^{2}-\frac{J_{b}}{\delta_{b2}^{2}}B_{2}^{2}\right)\left(S_{j}^{+}S_{j+1}^{-}+S_{j}^{-}S_{j+1}^{+}\right)\\ \; & +2\left(\frac{J_{a}^{2}}{\delta_{a1}^{3}}A_{1}^{2}+\frac{J_{b}^{2}}{\delta_{b1}^{3}}B_{1}^{2}\right)S_{j}^{Z}S_{j+2}^{Z}+\left(\frac{J_{a}^{2}}{\delta_{a2}^{3}}A_{2}^{2}-\frac{J_{b}^{2}}{\delta_{b2}^{3}}B_{2}^{2}\right)\left(S_{j}^{+}S_{j+2}^{-}+S_{j}^{-}S_{j+2}^{+}\right)+O\left(\frac{J_{\mu }^{3}}{\delta_{\mu i}^{3}}\right) \end{array}$$
for μ=a,b and i=1,2. The first two terms correspond to the NN interactions and the last two terms correspond to the NNN interactions. We can rewrite the above spin-chain Hamiltonian (with local magnetic fields) in a more compact form
(15)$$H=\sum_{j=1}^{N}\sum_{i=1}^{2}J_{i}\left(S_{j}^{X}S_{j+i}^{X}+S_{j}^{Y}S_{j+i}^{Y}\right)+\lambda_{i}S_{j}^{Z}S_{j+i}^{Z}+h_{j}S_{j}^{Z}$$
where
(16)$$\begin{array}{cc} J_{1} & =2\left(\frac{J_{a}}{\delta_{a2}^{2}}A_{2}^{2}-\frac{J_{b}}{\delta_{b2}^{2}}B_{2}^{2}\right)\\ \lambda_{1} & =2\left(\frac{J_{a}}{\delta_{a1}^{2}}A_{1}^{2}-\frac{J_{b}}{\delta_{b1}^{2}}B_{1}^{2}\right) \end{array}$$
are the NN interaction strengths, and
(17)$$\begin{array}{cc} J_{2} & =2\left(\frac{J_{a}^{2}}{\delta_{a2}^{3}}A_{2}^{2}+\frac{J_{b}^{2}}{\delta_{b2}^{3}}B_{2}^{2}\right)\\ \lambda_{2} & =2\left(\frac{J_{a}^{2}}{\delta_{a1}^{3}}A_{1}^{2}+\frac{J_{b}^{2}}{\delta_{b1}^{2}}B_{1}^{2}\right) \end{array}$$
are the NNN interaction strengths. Controlling the parameters appropriately, the MG model in Equation (Equation 9) can be realized.

The scheme cannot work without both $\Lambda_{a}$ and $\Lambda_{b}$. If we only consider the $\Lambda_{a}$ system, the NNN interaction strengths will be weaker than the NN interaction strengths by an order of J/δ. Since we require δ to be large, the NNN strengths will thus be very weak and cannot realize the MG Hamiltonian. With the two Λ systems, the driving phase can then produce the ${\left(-1\right)}^{j}$ factors in Equation (Equation 11), which can be utilized to suppress NN interactions and boost NNN interactions (notice the sign differences between Equation (Equation 16) and Equation (Equation 17) ). As such, the resultant NN and NNN interaction strengths can be made comparable by controlling the free laser and cavity parameters.

To obtain the MG Hamiltonian, we require $J_{1}=\lambda_{1}$, $J_{2}=\lambda_{2}$ and $J_{1}=2J_{2}$. This can be realized (for example) by setting $\delta_{1}=\delta_{2}$, $\Omega_{1}=\Omega_{3}$, $\Omega_{2}=\Omega_{4}$, $\Delta_{31}=\Delta_{32}$, $\Delta_{41}=\Delta_{42}$, $\delta_{31}=\delta_{32}$, $J_{a}=J_{b}$, $\delta_{a2}=\delta_{b2}$, $B_{2}=A_{2}/2$, $J_{a}=0.3\delta_{a2}$. As demonstrated in [26], the $J_{1}/J_{2}$ ratio can be adjust arbitrarily and can even be greater than 1, even though each term in $J_{1}$ is significantly larger than $J_{2}$. Moreover, the qubit is encoded in the ground states of the four-level atom, it is highly robust against decoherence due to dissipation. There is also negligible transitions to the excited states during the process due to large detunings, which justifies the adiabatic elimination of excited states.

Our scheme is experimentally feasible with state-of-the-art quantum devices such as microtoroidal cavities [27] and quantum dot-cavity systems [28]. The ground-state encoding of the qubit provides the robustness against radiative decay which thus leads to a long $T_{1}$ coherence time. The double Λ system can be realized with natural atoms such as the D1 line in cesium atoms, using the $6^{2}S_{1/2}$ hyperfine states as the ground states and the $6^{2}S_{3/2}$ hyperfine states as the excited states [29]. Multimode cavities (two cavity modes in our case) have also been experimentally demonstrated to achieve strong coupling with the atoms [30]. This is sufficient for our proposal as the effective coupling parameters, as given by the A and B parameters in our manuscript, can be increased (if necessary) by using a stronger laser driving field without requiring ultrastrong cavity-atom interactions.

## 4. Conclusions

We demonstrate the controllable quantum switchboard which directs the flow of quantum information in a quantum network, by allowing the sender to freely choose one of the two targets to receive the quantum state via teleportation. The key difference with other controlled teleportation protocols is that in the event of failure, the quantum information is not entirely lost and the switchboard behaves as an optimal telecloner, where the receivers can recover the quantum state with a maximal fidelity of $\frac{5}{6}$. This is achieved by preparing the sender, receivers, and controller in a four-qubit entangled state, which is necessary to function as both an optimal symmetric quantum cloning machine and deterministic controlled teleportation. We also provide an experimentally feasible physical implementation of the required four-qubit entangled state (as the ground state of the Majumdar–Ghosh model) using a coupled-cavity array. The quantum switchboard can potentially be a useful component of a large quantum network, where the information pathway can be redirected efficiently to reduce network congestion. The optimal telecloning aspect of our proposal provides a safeguard against the complete loss of quantum information in the worst-case scenario of a teleportation failure.

## Figures and Tables

**Figure 1 entropy-24-00136-f001:**
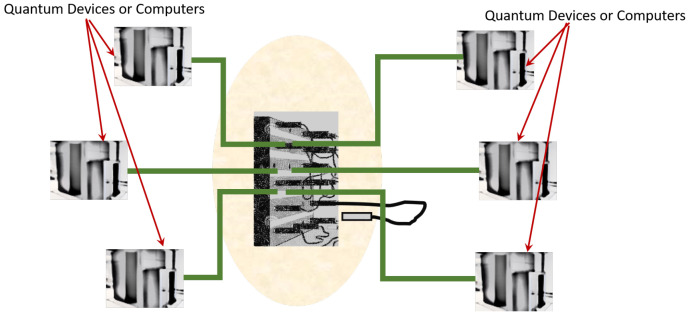
A quantum network consisting of many quantum devices and computers connected through a quantum switchboard. Switchboard can also redirect congested channels to less used channels during peak usage.

**Figure 2 entropy-24-00136-f002:**
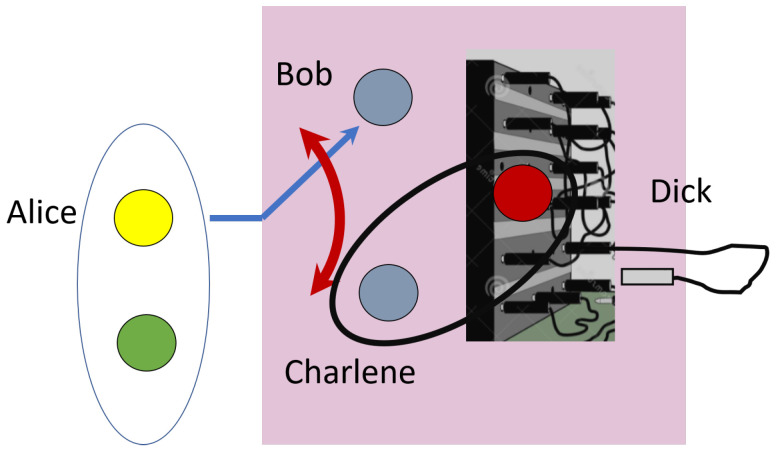
Schematic of quantum switchboard. Suppose Alice wishes to send her auxiliary qubit to Bob. She can direct Dick to send his qubit to Charlene (Bob). Charlene then performs a Bell measurement on his qubit with Dick’s qubit and sends results of his measurement to Bob. Using information from Charlene, Bob can perfectly recover state of Alice’s auxiliary qubit. Our switchboard state has unique feature of also being an optimal quantum cloning machine, which allows receivers to recover quantum state with maximal fidelity in event that teleportation protocol fails.

**Figure 3 entropy-24-00136-f003:**
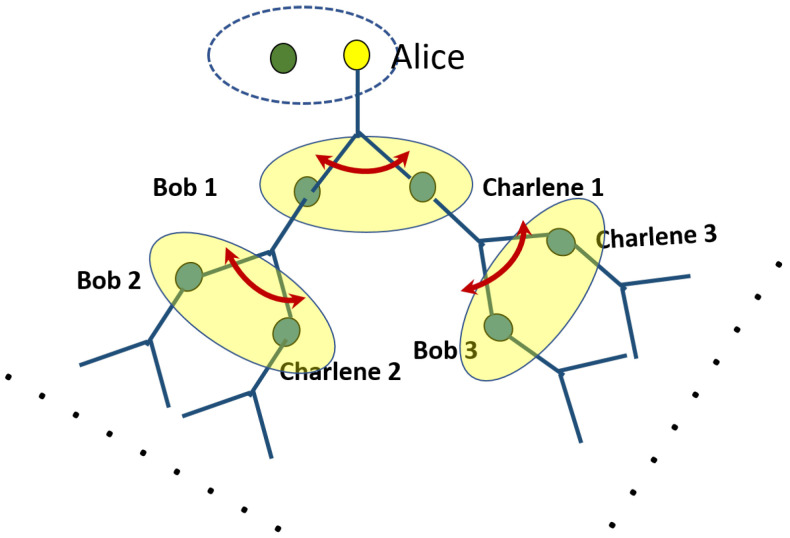
An illustration of how the scheme can be extended to a network of nodes with receivers labelled by Bob${}_{1}$, Bob${}_{2}$, ⋯ and Charlene${}_{1}$, Charlene${}_{2}$, ⋯. The controllers are not omitted from the diagram for simplicity, but are inherently present at each branch of the network to direct the flow of quantum information.

**Figure 4 entropy-24-00136-f004:**
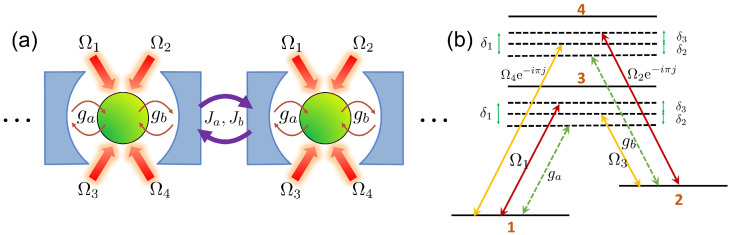
(**a**) Coupled-cavity array setup. Each unit cell contains a four-level atom coupled to a two-mode cavity field with strengths $g_{a}$ and $g_{b}$. Each atom is driven by four external lasers with Rabi frequencies $\Omega_{1},\Omega_{2},\Omega_{3}$ and $\Omega_{4}$. (**b**) Energy level diagram of the four-level atom. The detunings $\delta_{j}$ are defined in the main text. A double Λ system is formed, with $\Lambda_{a}=\left\{|1\rangle ,|2\rangle ,|3\rangle \right\}$ and $\Lambda_{b}=\left\{|1\rangle ,|2\rangle ,|4\rangle \right\}$. The ground states |1〉 and |2〉 are used to encode a qubit.

## Data Availability

Data sharing not applicable.

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
