# Peer review of "Quantum Switchboard with Coupled-Cavity Array"

_entropy, 2022, doi:10.3390/e24010136_

Round 1
Reviewer 1 Report
In this work, Mok et al. demonstrated a controllable quantum switchboard which directs the teleportation protocol to one of two targets, fully dependent on the sender’s choice. And they also provided a physical implementation of the proposal using a coupled-cavity array. The proposed switchboard can be utilized for the efficient routing of quantum information in a large quantum network.
The results seem correct and interesting. However, I have some points of criticism, which should be convincingly addressed by the authors.
- The symbol in Eq. (2) and the symbol below the Eq. (2) are not consistent.
- How do you get the fidelity 5/6, 1/3 and 2/3 in the section below the Eq. (5)?
- The formula in (10)shows incomplete.
Reviewer 2 Report
The authors, in the paper entitled “Quantum Switchboard with Coupled-Cavity Array”, demonstrate a controllable quantum switchboard which
directs the teleportation protocol to one of two targets, fully dependent on the sender’s choice. Quantum switchboard would be able to connect many quantum devices and computers in a quantum network, and also redirect congested channels to less used channels during peak usage.
The purpose of the work is unclear.
On the one hand the authors want to direct quantum information in a deterministic way towards two possible users, on the other hand they want to allow the solution of in the event that there is congestion of the channels.
The increase in resources needed to implement the idea is so high that any practical quantum communication infrastructure collapses. From a quantum engineering point of view, the robustness of the system is so low that an effective implementation would be dramatic even at small propagation distances.
The authors add the ability to telecloning, but the topic just feels like a filler.
Honestly, although the calculation is correct, the idea complicates the problem.
In this form, the paper can not be accepted for publication.
Reviewer 3 Report
I have difficulty in understanding equation (1). It seems a zero state, the two terms on the righthand of equation (1)are identical, namely (11)AB (11)DC and (11)AC(11)BD are the same., which means that there are four qubits in positions A, B, C, and D. Each qubit is in state |1>. Therefore, I could not proceed. I would like the authors to clarify this.
Round 2
Reviewer 3 Report
The revision is satisfactory. The protocol is presented clearly. I recommend its acceptance after minor revision. A related reference in directing information carrier flow in a network is also presented in ref. [1], which I suggest to include in a final revision.
Valagiannopoulos C. Optimized quantum filtering of matter waves with respect to incidence direction and impinging energy. Quantum Engineering, 2020, 2(3): e52.